# Sexually dimorphic neuronal responses to social isolation

**Laura Senst[1,2†], Dinara Baimoukhametova[1,2†], Toni-Lee Sterley[1,2], Jaideep Singh Bains[1,2*]**

[1]Department of Physiology and Pharmacology, University of Calgary, Calgary, Canada; [2]Hotchkiss Brain Institute, University of Calgary, Calgary, Canada

**Abstract** Many species use social networks to buffer the effects of stress. The mere absence of a social network, however, may also be stressful. We examined neuroendocrine, PVN CRH neurons and report that social isolation alters the intrinsic properties of these cells in sexually dimorphic fashion. Specifically, isolating preadolescent female mice from littermates for <24 hr increased first spike latency (FSL) and decreased excitability of CRH neurons. These changes were not evident in age-matched males. By contrast, subjecting either males (isolated or grouped) or group housed females to acute physical stress (swim), increased FSL. The increase in FSL following either social isolation or acute physical stress was blocked by the glucocorticoid synthesis inhibitor, metyrapone and mimicked by exogenous corticosterone. The increase in FSL results in a decrease in the excitability of CRH neurons. Our observations demonstrate that social isolation, but not acute physical stress has sex-specific effects on PVN CRH neurons.

*For correspondence: jsbains@ucalgary.ca

†These authors contributed equally to this work

Competing interests: The authors declare that no competing interests exist.

## Introduction

Survival threats trigger a repertoire of behavioral and endocrine adjustments (*Cannon, 1932*). The endocrine changes rely on the immediate engagement of the hypothalamic pituitary adrenal (HPA) axis which is controlled by corticotropin releasing hormone (CRH) neurons in the paraventricular nucleus of the hypothalamus (PVN). Although it is essential for survival, the dysregulation of the stress response is implicated in the emergence of numerous neuropsychiatric diseases (*Tost et al., 2015*). Interestingly, many of these psychopathologies show sex-specific differences which may be a consequence of sexual dimorphism of the neuroendocrine response to stress (*Bale and Epperson, 2015*). This sexual dimorphism may result in different response sensitivity to the same stressor and/or differences in which stimuli males and females perceive as stressful. Human studies demonstrating that young girls exhibit greater corticosteroid (CORT) stress reactivity to social stress tests than boys (*Gunnar et al., 2009*; *de Veld et al., 2012*) suggest, for example, that young females are more responsive to changes in social situations. Consistent with this idea, are demonstrations that, in comparison to males, females take greater advantage of social support and group dynamics to manage stress (*Taylor et al., 2000*). Here we hypothesized that disrupting the social network would elicit neurobiological changes preferentially in female mice.

## Results

To test this idea, mice were housed in same-sex groups (3–5 animals per group) from weaning (p21) until the day of the experiment (p22–35). This developmental window allows us to interrogate the contributions of sex differences independent of the effects of circulating gonadal hormones (*Nelson et al., 1990*). Mice were either in same-sex groups, pairs, or isolated from littermates for sixteen to eighteen hours. We then prepared hypothalamic brain slices and examined

**eLife digest** Many species, including humans, use social interaction to reduce the effects of stress. In fact, the lack of a social network may itself be a source of stress. Recent research suggests that young girls are more sensitive to social stress than boys. This could mean that social networks are more important for females in general, and that young females from different species, such as mice, may be more sensitive to social isolation than males. However, few studies have examined how social isolation affects the brain cells that control the release of stress hormones. As such, it remains unknown whether isolating individuals from their social group impacts on the brain in sex-specific ways.

Senst, Baimoukhametova et al. now show that the brains of young male and female mice react differently to social isolation. Less than a day after separation from their littermates, the activity in the brain cells of female mice became markedly different from that of isolated males. In contrast to social isolation, the physical stress of being made to swim produced similar changes in the brains of both male and female mice. Further experiments then showed that the changes in the brain cells that control the release of stress hormones required a signalling chemical called corticosterone, which is produced in response to stressful situations. This suggests that, in repsonse to soical isolation, the females are experiencing more stress than the males.

Following on from this work, one future challenge will be to investigate if reuniting a social group erases the effects of social isolation on the brain. Further experiments could also examine the behavioural and physiological effects of social isolation, including how females respond to later stressful events.

synaptic and intrinsic properties of identified CRH neurons in the PVN in acute brain slices (*Wamsteeker Cusulin et al., 2013b*) (*Figure 1a*). Activation of these cells is obligatory for launching the neuroendocrine response to stress (*Denver, 2009*).

The cellular heterogeneity in the PVN has necessitated various approaches to distinguish cell types. One of the most widely used has been to identify cells based on their voltage responses to a protocol of current steps. This reveals an 'electrical fingerprint' that has been used to distinguish among magnocellular (oxytocin, vasopressin) (*Tasker and Dudek, 1991*) and parvocellular (CRH, TRH, somatostatin) neurosecretory cells (*Hoffman et al., 1991*) and pre-autonomic neurons. Reports of changes in the intrinsic properties of neurons following experimental manipulations ex vivo (*Shah et al., 2010*; *O'Leary et al., 2014*; *Kourrich et al., 2015*), however, raise the possibility that electrical fingerprints are dynamic and potentially unreliable as a classification tool. We have recently described a transgenic reporter mouse in which CRH neurons can be identified and targeted directly for electrophysiological recording (*Wamsteeker Cusulin et al., 2013b*). In male mice, the electrical fingerprint of CRH cells was consistent with previous reports on parvocellular neurosecretory cells (*Wamsteeker Cusulin et al., 2013b*). These cells had a linear current voltage relationship and, unlike the magnocellular neurosecretory cells, did not have a long delay to first spike (first spike latency (FSL)) when depolarized from hyperpolarizing membrane potentials.

We first examined FSL of CRH neurons from group-housed vs isolated mice. FSL can play a key role in integrating synaptic events and influence probability of firing (*Molineux et al., 2005*) and changes in the underlying currents can tune neuronal frequency (*Ellis et al., 2007*). Social isolation had no effect on FSL in males (male$_{group}$: 48.7 $\pm$ 2.1 ms, n = 53 vs male$_{single}$: 46.1 $\pm$ 1.8 ms, n = 76, p=0.9; *Figure 1b,c*). By contrast, FSL in isolated females was significantly longer than FSL in group-housed females (female$_{group}$: 45.1 $\pm$ 1.9 ms, n = 42; female$_{single}$: 65.6 $\pm$ 1.9, n = 177, p<0.0001; *Figure 1b,d*). We then conducted experiments in which female mice were housed in pairs prior to electrophysiology experiments. This manipulation revealed an intermediate phenotype, with pair-housed females exhibiting FSLs that were longer than those observed in group housed female mice, but shorter than those observed in single-housed female mice (female$_{paired}$: 56.5 $\pm$ 2.0 ms, n = 105, p=0.0029 vs female$_{single}$ and p=0.0017 vs female$_{group}$, F$_{4,471}$ = 10.6). There was no difference in the input resistance of isolated males vs isolated females (male$_{single}$: 796 $\pm$ 31 MΩ, n = 157, vs female$_{single}$: 869 $\pm$ 23 MΩ, n = 273; unpaired t-test, p=0.063; *Figure 1—figure supplement 1*) or spike

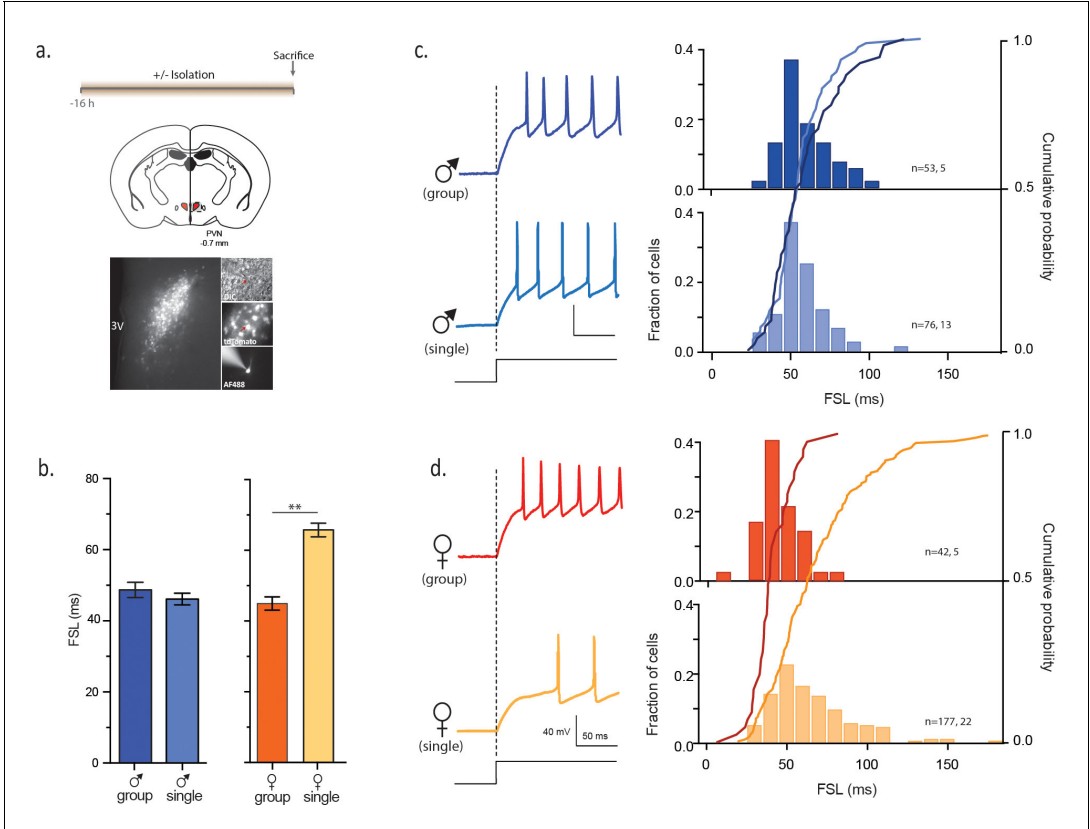

**Figure 1.** Social isolation alters intrinsic properties of CRH neurons in female, not male mice. (a) Experimental timeline (top), schematic (middle) showing coronal brain section of PVN (red) with a fluorescent image of tdTomato immunofluorescence (bottom). Panels on right show DIC image with electrode (top), tdTomato cells (middle) and recorded neuron (bottom) (b) Summary bar graphs indicate no difference in FSL for single and group-housed males (left, one-way ANOVA, p=0.9). FSL in single females is significantly longer than group-housed females (one-way ANOVA, p<0.0001). FSL in group-housed females and males are not different (p>0.05). (c) Traces show responses to a single +80 pA depolarizing step (holding potential = −102 mV) in group (dark blue) and single-housed (light blue) males. FSL calculated from the start of the depolarizing pulse (dotted line) to the first spike is plotted for all cells in (c). The relative frequency distributions are overlaid by the relative cumulative distributions. (d) Traces show responses to the same depolarizing step protocol as in (c) but in group (dark orange) and single-housed (light orange) females. The relative frequency distributions are overlaid by the relative cumulative distributions.

The following figure supplements are available for figure 1:

**Figure supplement 1.** No sex differences in basal properties of CRH neurons.

**Figure supplement 2.** No sex differences in excitatory post-synaptic currents.

**Figure supplement 3.** No sex differences in inhibitory post-synaptic currents.

threshold (male$_{single}$: −51.5 ± 1.0 mV, n = 63 vs female$_{single}$: −51.3 ± 0.7 mV, n = 99; unpaired t-test, p=0.86; *Figure 1—figure supplement 1*). We also examined properties of glutamate and GABA synapses on CRH neurons in hypothalamic slices from single-housed females and males. We observed no differences in basal glutamate (*Figure 1—figure supplement 2*) or GABA (*Figure 1—figure supplement 3*) synaptic transmission. These observations indicate that social isolation alters the intrinsic properties of CRH neurons in female, but not age-matched male mice.

Next, we hypothesized that female mice, but not male mice, may interpret social isolation as a stress. If true, we reasoned that subjecting single-housed male mice to an acute swim stress should also increase FSL. In females, however, the prior isolation should occlude the effects of an acute stress. To test this idea, single -housed male and female mice were subjected to swim stress for 20

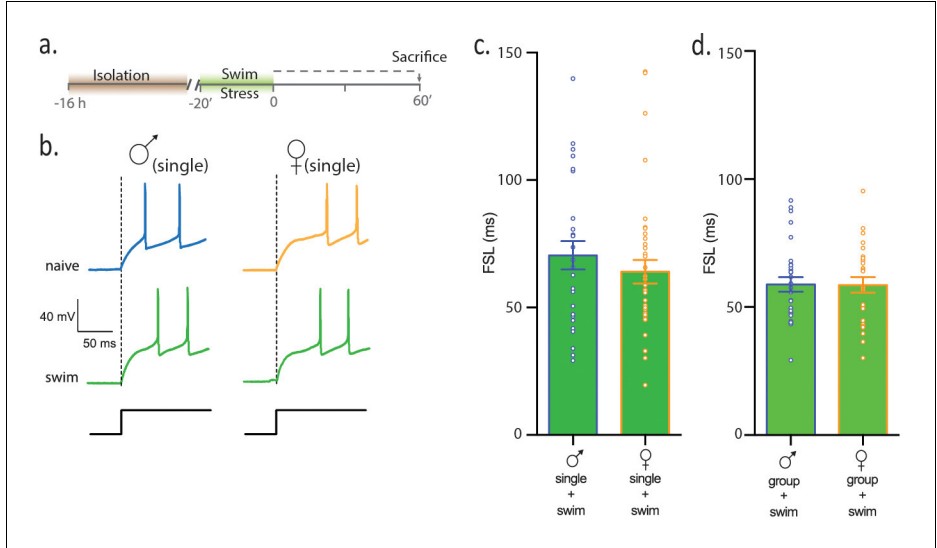

**Figure 2.** Females and males show equivalent sensitivity to an acute physical stress. (a) Experimental timeline. (b) Neuronal responses to +80 pA depolarizing pulse in single-housed males and females. (c) Summary data show that following swim stress, FSL in single males and single females is not different (p>0.05). FSL in single males subjected to swim is significantly longer than FSL in naive single males (p<0.0001). FSL in single females subjected to swim is not different than single females (p>0.05). (d) Summary data show that FSL in group-housed males and females is not different (p>0.05). Group-housed males subjected to swim have longer FSL than group-housed, naive males (p<0.001) and group-housed females subjected to swim have longer FSL than group-housed, naive females (p=0.0001).

min and intrinsic properties of CRH neurons were assessed (*Figure 2a*). Following swim stress, FSL in male mice (male$_{swim}$) was $64.0 \pm 4.6$ ms, n = 37, *Figure 2b,c*; this is significantly longer than FSL in the single males reported above (male$_{swim}$ vs male$_{single}$, 1 way ANOVA, p=0.0004). FSL in single female mice subjected to swim stress was $65.8 \pm 4.9$ ms, n = 32, *Figure 2b,c*; this is not different from FSL in female mice subjected to social isolation (female$_{swim}$ vs female$_{single}$, p=0.67). Finally, FSL in single males subjected to swim was not different from FSL in single females subjected to swim (p=0.9, *Figure 2b,c*). This suggests that either increases in FSL following social isolation occlude further effects of stress on CRH neurons or that females do not respond to swim stress. To test the effects of swim stress independent of any potential effects of social isolation, we conducted experiments in which group-housed females and males were subjected to swim stress. Following swim, FSL in group-housed males (male$_{group\ swim}$: $58.8 \pm 2.8$ ms, n = 23, *Figure 2d*) was significantly longer than FSL in naïve, group -housed males (p<0.001). FSL of group-housed males subjected to swim was not different than FSL in females subjected to swim (female$_{group\ swim}$:$58.7 \pm 1.9$ ms, n = 27, p>0.05 vs male$_{group}$, *Figure 2d*). Similarly, CRH neurons in group-housed females subjected to swim had a significantly longer FSL than CRH neurons in naive, group-housed females (p=0.0001). These observations demonstrate that both females and males show equivalent sensitivity to an acute physical stress (i.e. forced swim). Additionally, a prior social isolation occludes the effects of subsequent physical stress on FSL in females, but has no effect on FSL in males.

Since stress increases CORT, we examined the relationship between circulating CORT and FSL. In order to assess CORT during a defined temporal window, we subjected single male mice to swim stress and obtained samples thirty minutes after the protocol. We noted a positive correlation between plasma CORT and FSL ($r^2 = 0.3$, p=0.024, n = 17, *Figure 3a*). CORT, however, is one of many signaling molecules that is altered in response to stress. To determine whether CORT is necessary for increases in FSL following social isolation, we conducted experiments in which female mice were given access to drinking water containing the CORT-synthesis inhibitor, metyrapone prior to, and during, isolation (*Figure 3b*). Metyrapone treatment blocked the effects of social isolation on FSL in female mice (female$_{single\ metyrapone}$ = $47.2 \pm 2.0$ ms, n = 24, p=0.001 vs female$_{single}$,

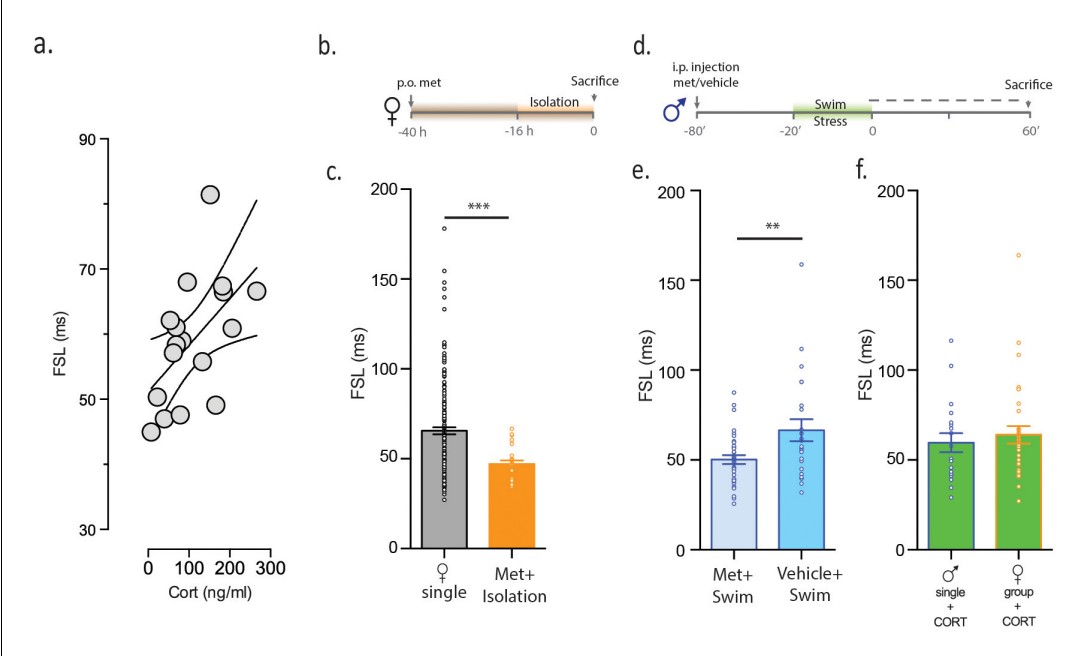

**Figure 3.** CORT is necessary and sufficient for increasing FSL. (a) CORT measurements in male mice subjected to stress are plotted against FSL. (b) Experimental timeline. (c) FSL following pre-treatment with the CORT synthesis inhibitor metyrapone in drinking water prior to and during isolation in females (orange). The gray bar shows data from *Figure 1* and is shown here for comparative purposes. (d) Experimental timeline. (e) Left bar shows FSL in male mice pretreated with Met prior to swim stress. Right bar shows FSL in male mice pretreated with vehicle prior to swim stress. (f) FSL is increased following incubation of slices from single males (p<0.021 vs male$_{single}$) or group-housed females (p=0.002 vs female$_{group}$) with CORT. There is no significant difference in FSL between the two CORT-treated groups.

*Figure 3c*). Next, to determine whether CORT is necessary for increases in FSL following swim stress, we administered metyrapone prior to swim stress in single males (*Figure 3d*). This eliminated the stress-induced increase in FSL in males (male$_{metyrapone}$ = 50.2 ± 2.4 ms, n = 35, vs male$_{vehicle}$ = 66.7 ± 2.4, n = 23, p=0.007, *Figure 3e*). These observations demonstrate that CORT is necessary for stress-induced changes in FSL in both males and females. In order to determine whether CORT is sufficient for increasing FSL, we incubated slices from either single-housed males or group-housed females with 100 nM CORT for 1 hr prior to electrophysiological assessment. This incubation time is sufficient to mimic CORT-dependent changes in synaptic metaplasticity observed after acute stress (*Wamsteeker Cusulin et al., 2013a*). CORT-incubation increased FSL in slices from single-housed males (59.2 ± 4.1 ms, n = 36, p<0.021 vs male$_{single}$) and in slices from group-housed females (64.1 ± 4.8 ms, n = 32, p<0.0001 vs female$_{group}$, *Figure 3f*). Collectively, these observations demonstrate that CORT is both necessary (during stress) and sufficient (in the absence of stress) to increase FSL in male and female mice.

Next we investigated the underlying conductance that controls FSL in single-housed males and females. A rapidly activating, rapidly inactivating potassium (K) conductance contributes to FSL in a number of different cell types throughout the nervous system (*Meng et al., 2011*). The channels responsible for this conductance are largely inactive at resting membrane potential, but this inactivation can be removed by a membrane hyperpolarization (*Yellen, 2002*; *Maffie and Rudy, 2008*). To test whether similar K channels contribute to the FSL in CRH neurons, we conducted experiments in which we did not deliver a membrane hyperpolarization prior to a depolarizing current step. The absence of this hyperpolarizing pre-pulse decreased the FSL in CRH neurons from socially-isolated females (FSL$_{noHP}$: 37.8 ± 3.1 ms vs FSL$_{HP}$: 55.3 ± 4.5 ms, paired t-test, p=0.0022, *Figure 4—figure supplement 1*). Rapidly activating and inactivating K currents are sensitive to millimolar concentrations of 4-aminopyridine (4-AP) (*Alexander et al., 2015*), and consistent with this, we observed shorter FSL in the presence of 2 mM 4-AP (FSL$_{4-AP}$: 36.8 ± 5.6 ms vs FSL$_{control}$: 62.8 ± 7.6 ms unpaired t-test, p=0.015 n = 8, *Figure 4a*). We obtained voltage clamp recordings and

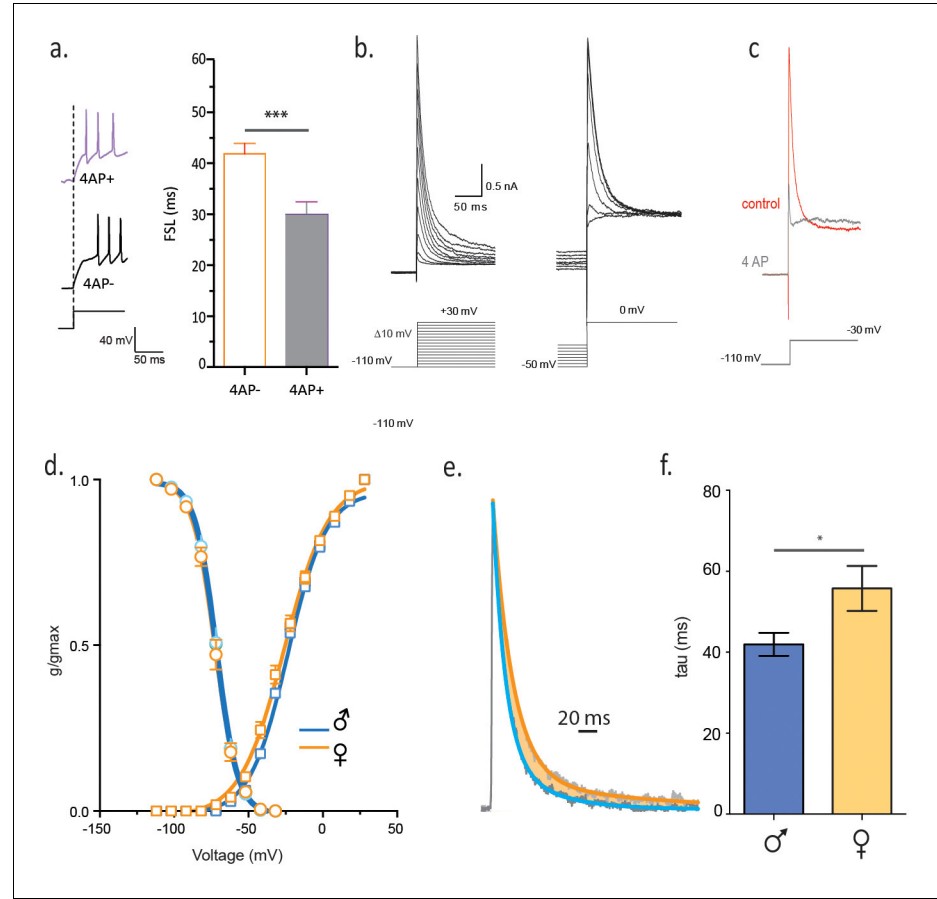

**Figure 4.** Slower decay of 4-AP sensitive, voltage-gated K current in isolated females. (**a**) FSL is decreased by bath application of 2 mM 4-AP. (**b**) Voltage clamp traces from a male CRH neuron in response to different current step protocols as shown. (**c**) The rapidly activating and inactivating current was completely blocked by 6 mM 4AP. (**d**) The activation and inactivation curves are shown for single-housed females and males. (**e**) Currents at half-maximal activation, fit with a double-exponential are shown from males and females. (**f**) The slow inactivation time constant in $male_{single}$ and $female_{single}$ is summarized (unpaired t-test, p=0.02).

The following figure supplement is available for figure 4:

**Figure supplement 1.** FSL is sensitive to membrane hyperpolarization prior to depolarization.

characterized the currents responsible for regulating FSL ($male_{single}$ n = 11, $female_{single}$ n = 12). These currents were largely inactive at resting membrane potential, inactivated quickly and required membrane hyperpolarization to relieve inactivation (*Figure 4b*). They were partially blocked by 2 mM 4-AP, fully blocked by 6 mM 4-AP (n = 4, *Figure 4c*), but were unaffected by a lower concentration of 4-AP(500 uM, data not shown) (*Anderson et al., 2010*). In addition, these currents were insensitive to 20 mM TEA (data not shown). Neurons from socially-isolated females showed no difference in activation or recovery from inactivation in comparison to isolated males (*Figure 4d*). There was no difference in current density in socially-isolated females when compared to socially-isolated males (data not shown), arguing against an increase in channel number or conductance. We next asked whether channel kinetics might explain the differences in FSL. There was no difference in the activation kinetics between males and females, but currents from female mice had a longer decay time constant in comparison to males ($\tau_{female\ single}$: 55.8 ± 5.6 ms, n = 11, vs $\tau_{male\ single}$: 41.9 ± 2.9 ms, n = 11, unpaired t-test, p=0.019, *Figure 4e,f*). These findings indicate that a slowing in the decay of a rapidly activating, rapidly inactivating voltage-gated K current is causative for the increase in FSL.

Next, we asked whether these differences in the biophysical properties of the rapidly inactivating K current observed in socially-isolated females compared to males would be sufficient to alter the spike output of CRH neurons. We conducted experiments in which depolarizing current steps (20 pA intervals) were preceded by a hyperpolarizing pre-pulse. CRH neurons from female isolated mice required a greater depolarization to generate a spike compared to CRH neurons from socially isolated males. This is revealed as a rightward shift in the relationship between spike probability and current step (female: $I_{50}$ = 49.5 ± 1.2 mV, n = 104 vs male: $I_{50}$ = 55.0 ± 1.6 mV, n = 88, p<0.0001, K-S statistic). Finally, we compared the excitability of CRH neurons from isolated females and males. We used a protocol (hyperpolarizing pre-pulse followed by a family of depolarizing steps) that allowed for maximal activation of the rapidly activating K current described above. The rightward shift in the F-I plot indicates that CRH neurons from isolated females had lower excitability in comparison to isolated males (p<0.0001, K-S statistic, *Figure 5b*).

## Discussion

Our findings demonstrate that brief social isolation (<24 hr) affects the biophysical properties of PVN CRH neurons in a sexually dimorphic fashion. We report an increase in FSL and a decrease in neuronal excitability in socially-isolated female, but not in age-matched, male mice. By contrast, an acute physical stress (swim) increased FSL in group-housed females and single-housed males. In addition, social isolation occluded the effects of swim stress on female mice. CORT is both necessary for the isolation (females) and swim stress (males and females) induced increases in FSL and sufficient to increase FSL in the absence of stress. Finally, we noted the effect of social isolation was graded in female mice with paired mice exhibiting FSLs that were intermediate to those observed from group-housed or single mice.

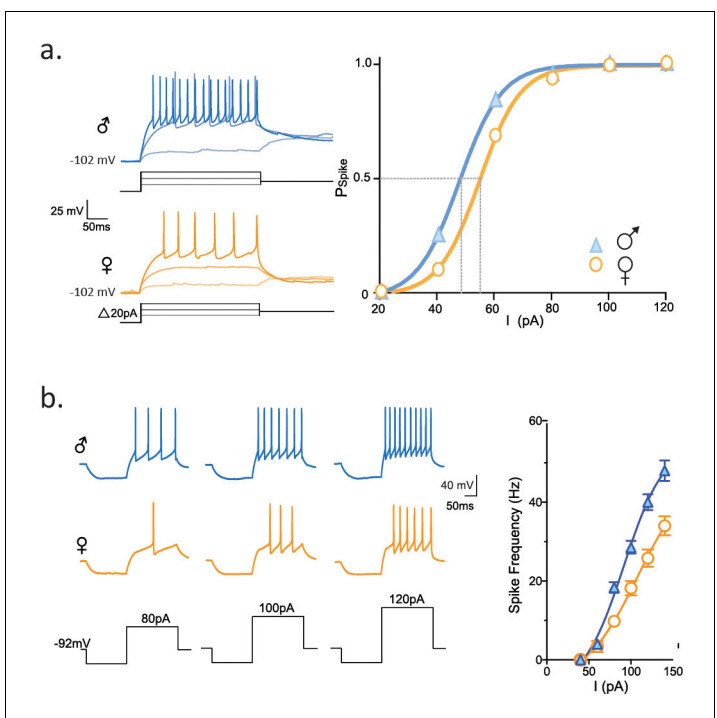

**Figure 5.** Longer FSL decreases spike probability and neuronal firing. (a) Traces show firing properties of CRH neurons from individually housed male (blue) and female (orange) mice. Right panel shows the probability of eliciting a spike in response to varying current steps (K-S statistic, p<0.0001). (b) Responses from male and female CRH neurons to varying current steps. Right panel shows spike frequency in the first 50 ms of current step (K-S statistic, p<0.0001).

We examined the underlying conductance and report that it is sensitive to millimolar concentrations of 4-AP and insensitive to millimolar concentrations of TEA. In addition, FSL was profoundly sensitive to pre-hyperpolarizing pulses, which remove channel inactivation and make rapidly inactivating K channels available for opening in response to depolarizations. The features of these channels are consistent with Kv4 channels previously described in the PVN (*Lee et al., 2012*). They do, however, exhibit a more hyperpolarized activation threshold in comparison to kinetically similar 4-AP sensitive channels reported in PVN parvocellular neurosecretory cells in rat (*Luther and Tasker, 2000*). We failed to note any differences in overall current density or voltage-dependent activation or recovery from inactivation. We did, however, observe an increase in the decay time constant of rapidly inactivating K currents in socially-isolated female mice compared to male mice, providing a potential underlying mechanism for the prolonged FSL in response to depolarization. The longer FSL we observed is not a classically defined characteristic of CRH neurons and, in fact, the absence of this delay has been used previously to categorize putative CRH neurons (*Hoffman et al., 1991*; *Wamsteeker Cusulin et al., 2013b*).

Our findings demonstrate that the biophysical characteristics of neurons are flexible and exquisitely sensitive to specific experiences. Whilst synaptic changes in the PVN (*Inoue et al., 2013*; *Kuzmiski et al., 2010*; *Wamsteeker Cusulin et al., 2013a*) and other brain regions have been investigated, this is, to the best of our knowledge, one of the first reports of rapid intrinsic plasticity in neurons following acute stress. Our observations add to a number of observations that intrinsic (synapse autonomous) plasticity can have a profound effect on the output of neural networks (*Shah et al., 2010*; *Kourrich et al., 2015*;*O'Leary et al., 2014*). In addition, as electrical fingerprints have been used to identify specific cell types, investigators should use caution when interpreting the data as the physiological or psychological state of the animal may have a dramatic impact on these fingerprints. Similarly, since neurons in the PVN can express multiple neuromodulators, caution must also be exercised when interpreting data from a genetically identified cell population. We have previously shown, using immunohistochemistry, that overlap between the genetically identified CRH cell population in PVN and non-CRH neurons is minimal (<5% for vasopressin/CRH or oxytocin/CRH; <<1% for thyrotropin releasing hormone/CRH or somatostatin/CRH) (*Wamsteeker Cusulin et al., 2013b*).

These findings demonstrate that males and females react differently to some stressors but not others. Specifically, social isolation has definitive, CORT-dependent effects on CRH neurons in female, but not male mice. By contrast, both males and females show indistinguishable responses to an acute physical swim stress. These findings highlight the importance of carefully considering both the sex of the animal and the stressor modality when designing studies investigating neurobiological effects of stress. Indeed recent work demonstrating sex-specific expression of fear responses in rodents (*Gruene et al., 2015*) serves as a corollary to our observations and when taken together suggests that strategies for coping with stress are sex-specific. In particular, our findings highlight the importance of a social network in females and provides a neurobiological framework for further testing the underlying thesis of 'tend and befriend' strategy for females in response to stress (*Taylor et al., 2000*). Finally, as our observations report sex-specific differences in response to stress in the first two weeks post-weaning, they provide clear evidence for biological effects on the stress axis that are independent of circulating gonadal hormones (*McCarthy and Arnold, 2011*). This exquisite sensitivity of the brain during the pre-adolescent period (*Foilb et al., 2011*) raises intriguing questions about the long-term consequences of subtle environmental and social manipulations on stress responses and behavior in males and females later in life.

## Materials and methods

### Animal care and stress paradigm

All protocols received approval from the University of Calgary Animal Care and Use Committee in accordance with the guidelines of the Canadian Council on Animal Care guidelines (Protocol # AC13-0027). B6(Cg)-Crh^tm1(cre)Zjh/J (*Crh-IRES-Cre*)mice and B6.Cg-Gt(ROSA)26Sor^tm14(CAG-TdTomato)Hze/J (*Ai14*) mice, whose generation has been detailed previously (*Madisen et al., 2010*; *Taniguchi et al., 2011*), were obtained from Jackson Laboratories (stock number 012704 and 007914 respectively). These were maintained as colonies of homozygous mice, with one

backcrossing to C57BL/6J background strain following their arrival. Genotyping was used to identify mutants using PCR procedures provided by the supplier. The following primers were used to identify *Crh-IRES-Cre* mutants: *5′-CTT ACA CAT TTC GTC CTA GCC* and *5′- CAA TGT ATC TTA TCA TGT CTG GAT CC-3′* and (468 base pair resultant PCR band). To identify *Ai14* mutants: *5′-GGC ATT AAA GCA GCG TAT CC-3′* and *5′-CTG TTC CTG TAC GGC ATG G -3′* were used (196 base pair band). The age of pre-adolescent mice (post-natal day 21–35) was determined according to previous literature demonstrating that in C57/BL mice the onset of puberty occurs at approximately 5 weeks of age (*Nelson et al., 1990*; *Mayer et al., 2010*). Mice were individually housed on a 12 hr:12 hr light: dark cycle (lights on at 7:00) with ad libitum access to food and water. Pairs of either homozygous *Crh-IRES-Cre* or *Ai14* genotypes were mated, and the resulting heterozygous *Crh-IRES-Cre;Ai14* offspring used in subsequent experiments. Sixteen hours prior to the acute stress protocol or slice preparation, mice were housed either individually, in same-sex pairs or same-sex groups (3–5 mice per group). Single or group -housed mice were randomly assigned to naive or stress conditions. For stress experiments, mice were exposed to a forced swim stress (between 8:00 and 9:30 during the light phase) consisting of 20 min in a glass cylinder (14 cm internal diameter) filled with 30–32°C water. Following one hour of recovery in their home cage, mice were anesthetized and brain slices were prepared as described below.

## Slice preparation

Experimental animals were anaesthetized with isoflurane and decapitated. The brain was quickly removed, and coronal brain slices (250 µm) containing the PVN of the hypothalamus were obtained using a vibrating slicer (Leica, Nussloch, Germany) while submerged in ice cold slicing solution (0°C, 95% $O_2$/5% $CO_2$ saturated), containing (in mM): 87 NaCl, 2.5 KCl, 0.5 $CaCl_2$, 7 $MgCl_2$, 25 $NaHCO_3$, 25 D-glucose, 1.25 $NaH_2PO_4$, 75 sucrose. Slices were then allowed a recovery period, of a minimum 60 min, in artificial cerebrospinal fluid (aCSF) (32.5°C, 95% $O_2$/5% $CO_2$ saturated) containing (in mM): 126 NaCl, 2.5 KCl, 26 $NaHCO_3$, 2.5 $CaCl_2$, 1.5 $MgCl_2$, 1.25 $NaH_2PO_4$, 10 glucose. The CORT synthesis inhibitor, metyrapone was dissolved in polyethylene glycol and injected i.p. 60 min prior to swim stress at a dose of 75 mg/kg in a volume of 50 µL. For female isolation experiments, CORT was dissolved in the drinking water (500 µg/ml) and given for 24 hr prior to isolation and during the entire isolation period.

## Electrophysiology

Hypothalamic slices were transferred to a recording chamber and superfused with 30–32°C aCSF at a flow rate of 1–2 ml/min. Slices were visualized using an upright microscope (BX51WI, Olympus) fitted with infrared differential interference contrast optics. CRH neurons were identified by their expression of tdTomato. Whole-cell patch clamp recordings were obtained from CRH neurons using borosilicate glass microelectrodes with tip resistance between 2–5 MΩ. The normal intracellular solution contained (in mM): 108 K-gluconate, 2 $MgCl_2$, 8 Na-gluconate, 8 KCl, 1 $K_2$-EGTA, 4 $K_2$-ATP, and 0.3 $Na_3$-GTP buffered with 10 mM HEPES. For images of filled cells, 0.2 mM Alexa-488 hydrazide and 10 mg·mL$^{-1}$ biocytin were added to internal solution. Microscope images were captured using a Retiga EXi camera (Qimaging) and processed using ImageJ.

Recordings were amplified using a Multiclamp 700B amplifier (Molecular Devices, Union City, CA), low-pass filtered at 1 kHz, and digitized at 10 kHz using the Digidata 1322 (Molecular Devices). Data were recorded (pClamp 9.2; Molecular Devices) and stored on a computer for offline analysis. During all experiments, initial access resistance ($R_a$) was below 20 MΩ. Cell membrane properties were monitored for the duration of the experiments and only recordings in which changes to $R_a$ $C_m$ did not exceed 15% were accepted for analysis. For synaptic experiments, the cell membrane was voltage clamped at −80 mV. Spontaneous inhibitory ionotropic $GABA_A$ receptor postsynaptic currents (sIPSCs) were isolated by blocking AMPA- and kainate-receptor-mediated glutamatergic synaptic transmission with 6,7-dinitroquinoxaline-2,3-dione (DNQX, 10 µM). When measuring spontaneous excitatory postsynaptic currents (EPSCs), the $GABA_A$ channel blocker picrotoxin (100 µM) was included in the bath to isolate excitatory currents mediated by AMPA and kainate receptors. These spontaneous currents representing stochastic transmitter release were analyzed using MiniAnalysis 6.0.3 (Synaptosoft, Decatur, GA). Event detection was set at three times the baseline noise and confirmed as synaptic events by eye.

To determine RMP, cells were recorded in zero current (I = 0) mode. Latency to spike initiation and firing thresholds were both measured using a current clamp depolarization step protocols. First, a baseline current injection that maintained the membrane voltage near −80 mV was chosen individually for each cell, which served to exclude the possible confounding influence of variable resting membrane potentials between cells. Next, a 200 ms/30 pA hyperpolarizing current injection was given followed by 250 ms/20 pA depolarizing current steps up to 140 pA. The latency time was measured as the duration from the point of initiating the depolarizing pulse to the initiation of the first spike. Firing threshold was determined as the membrane potential at the initiation of the first spike. Both firing threshold and membrane potential are corrected for a liquid junction potential of 12 mV, as calculated with solution ion concentrations.

In experiments examining cell properties following 4-AP administration, baseline recordings were obtained, and then a ten-minute treatment period was allowed before secondary data were obtained. Synaptic currents were evoked by paired afferent stimulation (every 5 s with an interstimulus interval of 50 ms) and analyzed using Clampfit 9.2 (Molecular Devices). Evoked postsynaptic current (ePSC) amplitudes were calculated from the baseline (current before the first evoked response) to peak of each evoked response. The paired pulse ratio (PPR) was calculated using the ratio of the amplitudes of the evoked pair (peak 2/peak 1) from a minimum of a one-minute epoch within each cell. To isolate the voltage gated fast inactivating K current the following cocktail was applied to the slices for at least of 15 minutes prior recording: Bicuculline 10 µM, DNQX 10 µM, dAPV 50 µM, TTX 1 µM, TEA 20 mM.

## Data analysis

Each group represents a minimum of three animals for pharmacology experiments, or a minimum of four for stress experiments. Data points are presented as mean ± SEM. Statistical analyses for sIPSCs, and current clamp step data were performed in GraphPad Prism 4 using a one way ANOVA for multiple groups followed by a post-hoc Tukey's multiple comparisons test; unpaired student's t-test were used for two group comparisons, and a K-S statistic for comparing two distributions.

## Drugs

Drugs were dissolved into aCSF daily prior to experiments from frozen aliquots stored at −20°C and added to the bath by perfusion pump. The drugs were dissolved in accordance with guidelines either in DMSO, PEG, ethanol or distilled water. DNQX, 4-AP, d APV and metyrapone were obtained from Tocris (Tocris Cookson, Ellisville, MO). Picrotoxin, bicuculline, TEA and corticosterone were obtained from Sigma (Sigma- Aldrich, St. Louis, MO). TTX was obtained from Alomone labs (Jerusalem BioPark (JBP), Hadassah Ein Kerem, P.O.Box 4287 Jerusalem 9104201, Israel).

## Acknowledgements

We thank all members of the Bains lab for thoughtful discussions and input. We are grateful to Ms. Cheryl Sank technical assistance and Mr. Rodney Barasi for animal husbandry. This work was supported by a CIHR Operating Grant (86501) to JSB. LS is supported by the Leaders in Medicine program in the Cumming School of Medicine. TLS is supported by a University of Calgary Eyes High Postdoctoral Fellowship and an Alberta Innovates Health Solutions Fellowship.

## Additional information

### Funding

| Funder | Grant reference number | Author |
|---|---|---|
| Leaders in Medicine, Cumming School of Medicine | | Laura Senst |
| Alberta Innovates - Health Solutions | Postdoctoral Fellowship | Toni-Lee Sterley |
| University of Calgary Eyes High Postdoctoral Fellowship | | Toni-Lee Sterley |

| Canadian Institutes of Health Research | 86501 | Jaideep Singh Bains |
| Fondation Brain Canada | | Jaideep Singh Bains |

The funders had no role in study design, data collection and interpretation, or the decision to submit the work for publication.

## Author contributions

LS, Conception and design, Acquisition of data, Analysis and interpretation of data, Drafting or revising the article; DB, T-LS, Acquisition of data, Analysis and interpretation of data, Drafting or revising the article; JSB, Conception and design, Analysis and interpretation of data, Drafting or revising the article

## Author ORCIDs

Jaideep Singh Bains, http://orcid.org/0000-0002-3634-6463

## Ethics

Animal experimentation: All experiments were approved by the University of Calgary Animal Care and Use Committee in accordance with Canadian Council on Animal Care guidelines: Protocol # AC13-0027.

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
