## [Decision Letter]

[Editors’ note: a previous version of this study was rejected after peer review, but the authors submitted for reconsideration. The previous decision letter after peer review is shown below.]

Thank you for submitting your work entitled "Sexually dimorphic neuronal responses to social isolation stress" for consideration by *eLife*. Your article has been favorably evaluated by a Senior Editor and four reviewers, one of whom, Peggy Mason, is a member of our Board of Reviewing Editors, and another is Danny Winder. Our decision has been reached after consultation between the reviewers. Based on these discussions and the individual reviews below, we regret to inform you that your work will not be considered further for publication in *eLife*.

The reviewers agreed that this manuscript addresses an important and understudied question regarding sex differences in the response of rodents to social isolation stress. Unfortunately, there were fundamental issues that decreased the reviewers' enthusiasm for this study.

1) Non-isolated animals were not studied. So the effect of social isolation on membrane properties of PVN CRH neurons is in fact not discernible from the experiments reported. That isolated females are different from isolated males may be due to sex, isolation, or an interaction of the two.

2) The metyrapone and RU486 experiments are done in males with FS instead of with isolated females. Also no vehicle controls.

3) Studied neurons may include neurons with up-regulated CRF transcript in response to isolation. Thus the population of neurons studied may not be as specific as suggested.

4) The observations reported are largely independent of each other and no causal relationship has been demonstrated.

The first two concerns, which were most critical to the editorial decision, are addressable with new experiments. Should the authors perform those experiments and substantially revise the manuscript to address the remaining concerns, then a new submission would be of interest.

*Reviewer #1:*

The fundamental problem with this study is that non-isolated animals are not studied. So the opening sentence of the discussion (that social isolation alters membrane properties of PVN CRH neurons in female rats) is simply not tested by the experiments. What can be said is that isolated females are different from isolated males. But whether that difference is due to sex, isolation or an interaction is unknown. Similarly, it is not known (although it is stated) that male PVN CRH neurons are unchanged by isolation.

The metyrapone and RU486 experiments were done on males and only males and only on FS. Why? There is no guarantee that the mechanisms of stress responses – when engaged – are the same in males and females.

*Reviewer #2:*

This manuscript starts off with an interesting and potentially important observation – that social isolation increases CRH PVN neuron first spike latency (FSL) in females, but not males. In addition, the authors nicely show that this effect can be partially reversed by pair housing (Figure 1). However, it's unclear to me why the authors then followed up by exploring corticosterone modulation of swim stress FSL increases in males and not females (Figure 2). Why didn't they look at CORT modulation of their initial finding (i.e. FSL in socially isolated females)? It therefore becomes difficult to directly link the findings in Figure 1 and Figure 2. Moreover, the experiments in Figure 3 also seem to be only loosely connected to the primary story. Based on the title of the paper, it is unclear why most of the mechanistic work (all of the CORT-related experiments) is done in males and not females, and the paper as a whole feels very disjointed, even if the individual experiments are more or less experimentally sound.

Additionally, there are a number of instances in which the manuscript or figure caption text is inconsistent or incomplete with respect to the figures. For example, "FSL in female mice subjected to swim stress was not different from FSL in female mice subjected to social isolation (Figure 2)," but Figure 2 compares female mice to male mice in the swim stress condition. Second, the manuscript text for Figure 3 refers to current characterization from voltage clamp recordings from male, female, and male + CORT groups, but traces are only shown from one male neuron, and the pink and blue traces are never identified (I'm assuming it's control and 4AP conditions, but this is not explicitly stated and the fact that the 4AP condition is represented in gray to the right makes things somewhat confusing). The leftward shift in the activation curve of socially isolated females is referenced in the manuscript as Figure 3, but this should be Figure 3.

*Reviewer #3:*

In this manuscript, Senst et al. explore sex-specific alterations in the excitability of CRF neurons in the hypothalamus. They present an interesting series of studies demonstrated a relationship between sex, social interaction and first spike latency (FSL) that is intriguing, and implicate SGK1 regulation of Kv4 potassium channels in the process. The work is timely, interesting, and well done, though the premise is slightly oversold. The second sentence of the manuscript states, "an unproven corollary is that the absence of a social network may itself be stressful". I would argue that this has already been widely studied and that "social isolation" is widely viewed as stressful. Less well characterized are the sex-specific differences in responses to this type of stress, which the authors nicely delve into here.

The presentation of comparisons is at first confusing, as the data seem to be presented in a counter-intuitive fashion. Rather than presenting data from group housed males and females, then comparing that with singly housed animals, the authors have presented the isolated data first. It would seem more logical and easier for the reader (at least this one) to follow to start with the "control" condition first in presentation, but this raises another issue, which is that it appears that group housed animals (beyond the paired) were not reported. This seems an important oversight to correct. That is, to demonstrate intrinsic plasticity has occurred by direct comparison of female group-housed to isolated, then to compare with males.

The description of the "paired-housed" experiments in the third paragraph of the Results is confusing, as it could be interpreted as implying the animals were isolated first, and then pair-housed. As such, does this represent a reversal of the plasticity? What if animals are only group housed? If such a "reversal" experiment was not performed, do the authors think subsequent pair-housing would reverse the isolation effects?

The authors show a correlation between cort levels and FSL in male mice. This initiates the mechanistic studies on GR and SGK1 relative involvement in FSL plasticity. However, little attention appears to be paid to the converse, that is, does the FSL plasticity observed lead to alterations in cort levels. This is a difficult "chicken-and-egg" issue, that nonetheless deserves more attention in the Discussion since the implicit idea is that this plasticity will lead to altered HPA output. The authors describe FSL plasticity as a means for negative regulation of HPA output, but since the latency is positively correlated with increased cort, this does not seem to be directly supported by the data.

*Reviewer #3 (Additional data files and statistical comments):*

Is the window current difference pointed to statistically significant? This needs to be explicitly described in the manuscript.

*Reviewer #4:*

This manuscript addresses an important and understudied question regarding sex differences in the response of rodents to social isolation stress. The authors show clear differences in the response of PVN CRF neurons to social isolation, with CRF neurons from females showing a marked decrease in excitability as measured by the latency to first spike onset (FSL) in response to a transient depolarizing current injection compared to male CRF neurons. The authors show that a similar response can be induced in males in response to an acute non-social stressor, suggesting that the response is sex- and stressor specific. In subsequent experiments the authors show that the increased FSL was mimicked by systemic cortisol administration, but was not blocked by a glucocorticoid receptor antagonist. In contrast, application of an inhibitor of serum glucocorticoid kinase-1 (SGK1) prevented the cortisol-induced increase in FSL. The authors then argue that mechanistically, the increased FSL, is due to a cort-SGK1-dependent modulation of an underlying Kv4 channel-mediated IA current.

The main concerns with the manuscript are that 1) the authors fail to address the potential confound of putative CRF neuronal heterogeneity in the PVN. Many cells in the brain transiently express mRNA for peptides but fail to express the mature protein, for example a subclass of oxytocin (OT) neurons in the PVN co-express CRF mRNA. Hence the transgenic mouse may show significant ectopic expression of the reporter protein used to "identify" CRF neurons. Significantly a subpopulation of OT neurons have properties identical to those of putative CRF neurons in the female PVN following isolation stress (Luther & Tasker, 2000), hence social isolation may upregulate CRF mRNA expression in these neurons and bias the sampling in subsequent recordings. 2) The authors fail to establish a cause-effect relationship for any of their independent observations. 3) In the absence of a dose-response relationship for 4-AP the assumption that the transient outward current is mediated by Kv4 channels is groundless. Many other transient outward K^+^ channels are blocked by lower concentrations of 4-AP. Also what evidence do the authors have that these cells even express Kv4 channels? 4) SGK1 activation modulates the activity of multiple ion channels that could affect the FSL including Kv1.1 – 1.5, M-channels, and ASIC1 channels, and the author provide no evidence for a direct/specific effect of SGK1 on Kv4 channel activity. Consequently, the manuscript reads like a collection of independent observations that have been rather tenuously linked together into a story that is mostly speculation.

[Editors’ note: what now follows is the decision letter after the authors submitted for further consideration.]

Thank you for submitting your article "Sexually dimorphic neuronal responses to social isolation" for consideration by *eLife*. Your article has been favorably evaluated by a Senior Editor and four reviewers, one of whom, Peggy Mason (Reviewer #1), is a member of our Board of Reviewing Editors, and another is Danny Winder (Reviewer #2 in the resubmission).

The reviewers have discussed the reviews with one another and the Reviewing Editor has drafted this decision to help you prepare a revised submission.

Your finding that isolation causes stress in females but not males is a striking and provocative example of sexual dimorphism. Yet the follow up experiments do not convincingly investigate this sex difference since, for example, there are no non-isolated control groups. Also without female CORT data or swim stress in group-housed animals, the sexual dimorphism is restricted to whether isolation is sufficient or not.

On the other hand, the mechanistic insight into the stress response, afforded by the FSL result, is exciting. The authors are urged to change the emphasis (and title) accordingly.

*Reviewer #1:*

This rewrite is very well done. The experiments are well motivated and clear in the revision. I have only a few comments.

Two portions of the text are a bit challenging to follow. First the contribution of the rapidly inactivating K channels. This is a broad audience. Please remind the reader what the meaning of each finding is and how the results lead step by step to the conclusion.

Second, the negative feedback part is confusing to me. I was happily going along thinking that I was learning about the stress response to social isolation (F) or swim in cold water (M) and then in the Discussion I read that the whole paper is interpreted as the negative feedback on the stress response. Obviously, I, as is true of most readers, do not think about stress responses as much as the authors do. The authors are requested to take this into account and connect the dots. Is the stress response so fleeting that any attempt to study it is thwarted and automatically delegated to the longer-lasting negative feedback stage?

In the first experiment on group- vs. single-housed mice, what age are these animals? Are these adults? If not, when in the P22-35 window do these animals come from? Were there any differences across this broad range of isolation times?

In the Abstract, it should be stated that exposure of male mice to acute *physical* stress had the same effect.

The authors may consider rephrasing the conclusion that stress is perceived differently by males and females. This reviewer suggests: males and females react differently to stress.

Why was Metyrapone administered p.o. to females and by injection to males?

Interesting questions for the future include: Are females similarly sensitive to physical stressors as are males? Would grp housed females show shortened FSL after a swim? The authors could consider adding in such speculation as their Discussion is laudably direct and concise.

*Reviewer #2:*

This revised version is very significantly improved over the initial submission. I have no significant concerns to add.

*Reviewer #3:*

Overall I think this is an improved manuscript. Figure 1 and Figure 3 are more complete, and I think the significance is high. However, I still find concerning gaps in the connectivity of the individual experiments and the conclusions drawn.

I am still confused by the logic behind the 2nd set of experiments. If the hypothesis is that FSL changes in isolated females are due to stress, then how does stressing isolated males test this hypothesis? (Results, fourth paragraph). It seems to me that it could just be a matter of degree – i.e. isolation isn't sufficiently stressful to increase FSL in males, but isolation + swim stress is. From my reading, the authors did not test swim stress alone in either group. Conversely, the lack of FSL increase in response to swim stress in isolated females seems more like a ceiling effect than an occlusion. To be able to conclude that the isolation-induced FSL increase in females is due to stress, the authors should be able to show the same effect in group housed females exposed to swim stress. Additionally, Figure 2 shows the CORT/FSL correlation for males only, when the primary hypothesis seems to be for females. If this correlation wasn't significant in females, that should be stated. To me, these were the missing experiments and data sets in the first version of the paper and they appear to still be missing.

I can't find the data described at the end of the fourth paragraph of Results (CORT effects in vitro). First, why wouldn't the authors show these data, and second, why does it appear that this experiment was only done in isolated males? Response point 6 says that they didn't do it in females because FSL is already increased due to isolation, but they could have done it in non-isolated animals to show that CORT can have an effect on FSL in females. As with the first version of this manuscript, these kinds of decisions make it seem like the authors are throwing together a bunch of data without considering how each experiment contributes to the overarching conclusions. If the goal of this project is to define sexual dimorphism in CRH neuronal responses to stress and stress hormones, they need to do these experiments in both sexes and experimentally dissect out the roles of isolation vs. swim stress, since in many experiments these conditions are combined.

*Reviewer #4:*

The authors have gone a long way in addressing the concerns raised in the initial review of this revised manuscript. The addition of the non-stressed control groups, and the metyrapone experiments in females have strengthened the manuscript considerably. Similarly, the removal of the SGK1 data and the inclusion of the CORT experiments in females have bolstered the mechanistic aspect of the study. Consequently, I have no major concerns with the manuscript. The study is timely, and the identification of sex-specific responses of the PVN circuitry to stressors is of considerable interest.

---

## [Author Response]

[Editors’ note: the author responses to the first round of peer review follow.]

*The reviewers agreed that this manuscript addresses an important and understudied question regarding sex differences in the response of rodents to social isolation stress. Unfortunately, there were fundamental issues that decreased the reviewers' enthusiasm for this study.*

*1) Non-isolated animals were not studied. So the effect of social isolation on membrane properties of PVN CRH neurons is in fact not discernible from the experiments reported. That isolated females are different from isolated males may be due to sex, isolation, or an interaction of the two.*

We have now conducted additional experiments with male and female animals housed in same-sex groups (3-5 mice per group). There is no difference in first spike latency (FSL) if males are isolated or in a group. By contrast, females show an inverse correlation between group size and FSL. FSL increases as we decrease from group housing (3-5) to pair housing to single housing. These data are now part of a new Figure 1.

*2) The metyrapone and RU486 experiments are done in males with FS instead of with isolated females. Also no vehicle controls.*

The metyrapone experiments have now been done in females as follows: Females in a group (3) were given access to drinking water containing metyrapone for one day and then isolated for sixteen hours with continued access to metyrapone in the water. The FSL in CRH neurons recorded from these animals was not different from group-housed females or from males pre-treated with metyrapone and then subjected to swim stress. These data are now reported in Figure 3.

We have also conducted vehicle control experiments. Vehicle (PEG) administration had no effect on the stress-induced increase in FSL. For the female isolation experiments, metyrapone was dissolved in drinking water, so the control group is the isolated females.

*3) Studied neurons may include neurons with up-regulated CRF transcript in response to isolation. Thus the population of neurons studied may not be as specific as suggested.*

This is a point that can be raised with every study that uses a genetic approach to target a specific cell population. The alternatives to this approach are to use post-hoc immunohistochemistry to identify the cell. This is notoriously unreliable and in particular, immunohistochemistry for CRH rarely labels more than 75% of the cell population. One could use single-cell PCR approaches, but here the reviewer’s concerns would be the same. There is no perfect method, but we have carefully quantified potential overlap with other cell populations in PVN in a previous paper (Wamsteeker Cusulin et al., 2013). We see less than 5% overlap with oxytocin neurons and td-tomato in PVN. More than 95% of the CRH immunopositive neurons are td tomato positive. We do agree with the reviewer that some caution is warranted, so we have added a comment in the Discussion to reflect this point.

*4) The observations reported are largely independent of each other and no causal relationship has been demonstrated.*

Our additional experiments with group housed females and males as well as those showing the effects of isolation on FSL were blocked by inhibition of CORT synthesis strengthen the links between the findings. In addition, we have taken the reviewer comments about the flow and logic of the manuscript very seriously and modified the organization of the text and figures accordingly. In summary, we now show that:

1) Social isolation increases FSL in females, but not males.

2) The impact of social isolation was blocked by administering the CORT-synthesis inhibitor, metyrapone to females prior to and immediately after social isolation. This suggested that CORT was necessary for the increase in FSL.

3) We then asked whether acute stress, which increases CORT, would similarly increase FSL in males. It did.

4) If the increase in FSL in socially-isolated females is due to CORT, then effects of additional swim stress should be occluded. They were.

5) We show a positive correlation between CORT and FSL.

6) To determine whether CORT is sufficient, we incubated naïve males slices with CORT for 60 minutes and showed that this increased FSL. Female slices were not used because FSL is already increased in socially-isolated females.

7) Mechanistically, this increase in FSL relies on a Kv channel that is sensitive to millimolar concentrations of 4-aminopyridine. There was no change in current density after stress, but we do report significantly longer decay kinetics in female CRH neurons.

8) The functional consequences of this intrinsic plasticity is a decrease in excitability of CRH neurons from socially isolated females.

*The first two concerns, which were most critical to the editorial decision, are addressable with new experiments. Should the authors perform those experiments and substantially revise the manuscript to address the remaining concerns, then a new submission would be of interest.*

*Reviewer #1:*

*The fundamental problem with this study is that non-isolated animals are not studied. So the opening sentence of the discussion (that social isolation alters membrane properties of PVN CRH neurons in female rats) is simply not tested by the experiments. What can be said is that isolated females are different from isolated males. But whether that difference is due to sex, isolation or an interaction is unknown. Similarly, it is not known (although it is stated) that male PVN CRH neurons are unchanged by isolation.*

We have conducted additional experiments to address this question directly. Specifically, males and females were housed in same sex groups (3-5 animals per group). This manipulation had no effect on FSL in males (compared to single housed), but there was a significant decrease in FSL in females (compared to single-housed). The data from these experiments are now shown in a new Figure 1. Additional experiments in which females were pair-housed showed an intermediate FSL phenotype. Collectively, these observations indicate a graded effect of the social network in females that is absent in males. In order to make the study easier to understand, we have re-written the entire manuscript, re-organized the figures and conducted additional experiments to address reviewer concerns.

*The metyrapone and RU486 experiments were done on males and only males and only on FS. Why? There is no guarantee that the mechanisms of stress responses – when engaged – are the same in males and females.*

We conducted these experiments in males because single-housed females already showed an increase in FSL, and therefore additional effects of swim stress were occluded. We have now conducted additional metyrapone experiments in the females (Figure 3). Given the additional experiments that have significantly strengthened the link between social isolation and changes in FSL, we feel the RU486 experiments divert attention from the main hypothesis and consequently, we have removed them from the manuscript.

*Reviewer #2:*

*This manuscript starts off with an interesting and potentially important observation – that social isolation increases CRH PVN neuron first spike latency (FSL) in females, but not males. In addition, the authors nicely show that this effect can be partially reversed by pair housing (Figure 1). However, it's unclear to me why the authors then followed up by exploring corticosterone modulation of swim stress FSL increases in males and not females (Figure 2). Why didn't they look at CORT modulation of their initial finding (i.e. FSL in socially isolated females)? It therefore becomes difficult to directly link the findings in Figure 1 and Figure 2. Moreover, the experiments in Figure 3 also seem to be only loosely connected to the primary story. Based on the title of the paper, it is unclear why most of the mechanistic work (all of the CORT-related experiments) is done in males and not females, and the paper as a whole feels very disjointed, even if the individual experiments are more or less experimentally sound.*

As described in the response to the editor, we have now conducted metyrapone experiments in females prior to and immediately after social isolation. This eliminated the increase in FSL. These data are now included in Figure 3. The CORT modulation was done in males because an acute stress in males mimicked the effects of social isolation in females. This hinted at a common mechanism. Furthermore, the fact that acute stress had no effect on FSL in females suggests that isolation has occluded the effects of acute elevations of CORT.

We have thought carefully about the reviewer’s comment about the ‘disjointed’ aspects of the paper and the fact that experiments in Figure 3 appear only loosely connected. Accordingly, and because of new data, we have re-organized the paper and removed the data in Figure 3 because we agree that they were largely diversionary.

*Additionally, there are a number of instances in which the manuscript or figure caption text is inconsistent or incomplete with respect to the figures. For example, "FSL in female mice subjected to swim stress was not different from FSL in female mice subjected to social isolation (Figure 2)," but Figure 2 compares female mice to male mice in the swim stress condition. Second, the manuscript text for Figure 3 refers to current characterization from voltage clamp recordings from male, female, and male + CORT groups, but traces are only shown from one male neuron, and the pink and blue traces are never identified (I'm assuming it's control and 4AP conditions, but this is not explicitly stated and the fact that the 4AP condition is represented in gray to the right makes things somewhat confusing). The leftward shift in the activation curve of socially isolated females is referenced in the manuscript as Figure 3, but this should be Figure 3.*

We apologize for these errors. We have corrected the manuscript accordingly.

*Reviewer #3:*

*In this manuscript, Senst et al. explore sex-specific alterations in the excitability of CRF neurons in the hypothalamus. They present an interesting series of studies demonstrated a relationship between sex, social interaction and first spike latency (FSL) that is intriguing, and implicate SGK1 regulation of Kv4 potassium channels in the process. The work is timely, interesting, and well done, though the premise is slightly oversold. The second sentence of the manuscript states, "an unproven corollary is that the absence of a social network may itself be stressful". I would argue that this has already been widely studied and that "social isolation" is widely viewed as stressful. Less well characterized are the sex-specific differences in responses to this type of stress, which the authors nicely delve into here.*

We have re-written parts of the Introduction to make this point clearer.

*The presentation of comparisons is at first confusing, as the data seem to be presented in a counter-intuitive fashion. Rather than presenting data from group housed males and females, then comparing that with singly housed animals, the authors have presented the isolated data first. It would seem more logical and easier for the reader (at least this one) to follow to start with the "control" condition first in presentation, but this raises another issue, which is that it appears that group housed animals (beyond the paired) were not reported. This seems an important oversight to correct. That is, to demonstrate intrinsic plasticity has occurred by direct comparison of female group-housed to isolated, then to compare with males.*

We have re-organized the Results to start with the ‘control’ condition (group-housed males or females) and now include additional experiments with metyrapone in females (see comments to editor and reviewer 1 above).

*The description of the "paired-housed" experiments in the third paragraph of the Results is confusing, as it could be interpreted as implying the animals were isolated first, and then pair-housed. As such, does this represent a reversal of the plasticity? What if animals are only group housed? If such a "reversal" experiment was not performed, do the authors think subsequent pair-housing would reverse the isolation effects?*

The animals were not isolated first. They were either in pairs or single. We did not attempt a reversal experiment. Given the addition of new data with males or females in groups, the paired female data, showing an effect that is intermediate, now serve to further support our overall argument.

*The authors show a correlation between cort levels and FSL in male mice. This initiates the mechanistic studies on GR and SGK1 relative involvement in FSL plasticity. However, little attention appears to be paid to the converse, that is, does the FSL plasticity observed lead to alterations in cort levels. This is a difficult "chicken-and-egg" issue, that nonetheless deserves more attention in the Discussion since the implicit idea is that this plasticity will lead to altered HPA output. The authors describe FSL plasticity as a means for negative regulation of HPA output, but since the latency is positively correlated with increased cort, this does not seem to be directly supported by the data.*

We have now conducted experiments in which we block CORT production in females prior to isolation. Following this manipulation, FSLs are the same as those observed in group housed females or males. We have removed the SGK data because we agree with a comment raised by reviewer 2 that these were diversionary. Instead, we have bolstered the link between isolation, stress, CORT and FSL.

*Reviewer #3 (Additional data files and statistical comments):*

*Is the window current difference pointed to statistically significant? This needs to be explicitly described in the manuscript.*

We have re-analyzed the window current data, and although the curve-fits appear to show differences, we are not convinced of its biological relevance given that the curves do not always fit the data points accurately. Consequently, we are no longer making any statements about window current differences. We have conducted additional analyses on the kinetics of the fast-inactivating current in males and females. We report that currents from isolated females have significantly longer time constants (Figure 4). Given the robust increase in FSL and the absence of any changes in current density and only modest changes in inactivation/activation thresholds, we propose that kinetic changes in this current are likely responsible for the longer FSL in isolated females.

*Reviewer #4:*

*The main concerns with the manuscript are that 1) the authors fail to address the potential confound of putative CRF neuronal heterogeneity in the PVN. Many cells in the brain transiently express mRNA for peptides but fail to express the mature protein, for example a subclass of oxytocin (OT) neurons in the PVN co-express CRF mRNA. Hence the transgenic mouse may show significant ectopic expression of the reporter protein used to "identify" CRF neurons. Significantly a subpopulation of OT neurons have properties identical to those of putative CRF neurons in the female PVN following isolation stress (Luther & Tasker, 2000), hence social isolation may upregulate CRF mRNA expression in these neurons and bias the sampling in subsequent recordings.*

We have previously shown that 96% of the CRH immunopositive neurons in PVN are also tdTomato positive (Wamsteeker Cusulin et al., 2013). These cells are neuroendocrine (terminals project outside the blood-brain barrier) as 86% of td-Tomato cells are labelled by intravenous (tail vein) injection of fluorogold. There is virtually no overlap (3%) between OT neurons and td-Tomato neurons; 5% overlap between VP and td-Tomato and <<1% overlap between either TRH (3 cells in 5 mice) or somatostatin (0 cells in 5 mice).

The reviewer’s astute comment, however, raises a larger issue that can, and perhaps should, be raised with every manuscript that uses a genetic approach to identify neurons. That concern is about how we actually describe or categorize neurons. If a neuron does not make a peptide, for example, CRH when sampled, does that mean it’s not a CRH neuron? We did not set out to resolve this question and we do not think anyone has resolved this in a satisfactory fashion. If the reviewer can offer a better alternative of sampling CRH neurons, we’re open to suggestions. Second, we are well aware of the Luther and Tasker paper (Luther & Tasker, 2000) and in fact it is that paper (and others) that have led the field to the conclusion that there are distinct fingerprints that can be used to distinguish OT/VP neurons from non-magnocellular neuroendocrine cells (CRH, TRH, somatostatin) and preautonomic neurons. We are arguing here that it is not so simple and that the plasticity of intrinsic properties is something that should be considered.

*2) The authors fail to establish a cause-effect relationship for any of their independent observations. 3) In the absence of a dose-response relationship for 4-AP the assumption that the transient outward current is mediated by Kv4 channels is groundless. Many other transient outward K^+^ channels are blocked by lower concentrations of 4-AP. Also what evidence do the authors have that these cells even express Kv4 channels?*

The reviewer makes a good point. Based on pharmacological overlap, it is difficult to discriminate between specific Kv channel subtypes. At this stage, based on our ability to block >30% of the current at 1-2 mM 4-AP, we can only conclude that these are either Kv1.4 or Kv4. We choose these as candidates based on work from Lee et al. indicating that Kv1.2, Kv1.3, Kv1.4, Kv4.1, Kv4.2 are expressed in neuroendocrine cells in PVN(Lee et al., 2012). We had shown this in the manuscript. Kv1.2 and 1.3 exhibit characteristics of classic Kv channels with delayed activation and no inactivation and are not sensitive to 4-AP, so we can rule these out as candidates. Kv1.4 and Kv4 are sensitive to 4-AP (fampridine), therefore we cannot discriminate between these options. Consequently, we have not suggested a specific K channel subtype and instead, refer to them transient outward K^+^ channels.

*4) SGK1 activation modulates the activity of multiple ion channels that could affect the FSL including Kv1.1 – 1.5, M-channels, and ASIC1 channels, and the author provide no evidence for a direct/specific effect of SGK1 on Kv4 channel activity.*

Based on comments from reviewer 2, we have removed the SGK1 data from the manuscript.

[Editors’ note: the author responses to the re-review follow.]

Your finding that isolation causes stress in females but not males is a striking and provocative example of sexual dimorphism. Yet the follow up experiments do not convincingly investigate this sex difference since, for example, there are no non-isolated control groups.

Non-isolated control groups are shown in Figure 1, Figure 2 and Figure 3.

*Also without female CORT data or swim stress in group-housed animals, the sexual dimorphism is restricted to whether isolation is sufficient or not.*

We have also conducted additional experiments in which group housed female mice were subjected to swim stress. As expected, this increased FSL. We also conducted experiments in which slices from group housed female mice were incubated with CORT. This increased FSL. The results are consistent with our model and more detailed information is available in specific responses to reviewer 2.

*Reviewer #1:*

*This rewrite is very well done. The experiments are well motivated and clear in the revision. I have only a few comments.*

*Two portions of the text are a bit challenging to follow. First the contribution of the rapidly inactivating K channels. This is a broad audience. Please remind the reader what the meaning of each finding is and how the results lead step by step to the conclusion.*

This is a very good point and we have made changes throughout the manuscript to provide a succinct conclusion at the end of each experiment, followed by a sentence that provides the rationale for the next series of experiments.

*Second, the negative feedback part is confusing to me. I was happily going along thinking that I was learning about the stress response to social isolation (F) or swim in cold water (M) and then in the Discussion I read that the whole paper is interpreted as the negative feedback on the stress response. Obviously, I, as is true of most readers, do not think about stress responses as much as the authors do. The authors are requested to take this into account and connect the dots. Is the stress response so fleeting that any attempt to study it is thwarted and automatically delegated to the longer-lasting negative feedback stage?*

We apologize for this lack of clarity. We have made a number of changes to the discussion which are in blue. We are in agreement with the reviewer that the negative feedback discussion is confusing (specifically for a broad audience) and have removed this from the manuscript.

*In the first experiment on group- vs. single-housed mice, what age are these animals? Are these adults? If not, when in the P22-35 window do these animals come from? Were there any differences across this broad range of isolation times?*

We did examine the effects of age on FSL in socially isolated females. There were no significant differences across this range (p=0.23) (see Figure 6).

Author response image 1.**DOI:**
http://dx.doi.org/10.7554/eLife.18726.012

*In the Abstract, it should be stated that exposure of male mice to acute physical stress had the same effect.*

Done.

*The authors may consider rephrasing the conclusion that stress is perceived differently by males and females. This reviewer suggests: males and females react differently to stress.*

Done.

*Why was Metyrapone administered p.o. to females and by injection to males?*

Metyrapone has relatively short half-life and the effects of the injection are rapid, but transient. So, injection was appropriate prior to swim stress in the males, but because we did not know when, during isolation, CORT peaked, we decided to administer metyrapone through the drinking water in the females to maintain a constant dose throughout the experiment.

*Interesting questions for the future include: Are females similarly sensitive to physical stressors as are males? Would grp housed females show shortened FSL after a swim? The authors could consider adding in such speculation as their Discussion is laudably direct and concise.*

Since reviewer 3 also suggested a similar experiment, we have now done this and include the data in the manuscript. Group-housed females were subjected to swim stress. This increased FSL (see response to reviewer 2 for additional details). FSL in males subjected to swim stress was not different from FSL in females subjected to swim or social isolation. So, one additional conclusion we can draw is that physical stressors have the same effect on males and females, but the response to social isolation is sexually dimorphic. We have now expanded the Discussion according to the suggestions provided here.

*Reviewer #3:*

*Overall I think this is an improved manuscript. Figure 1 and Figure 3 are more complete, and I think the significance is high. However, I still find concerning gaps in the connectivity of the individual experiments and the conclusions drawn.*

*I am still confused by the logic behind the 2nd set of experiments. If the hypothesis is that FSL changes in isolated females are due to stress, then how does stressing isolated males test this hypothesis? (Results, fourth paragraph). It seems to me that it could just be a matter of degree – i.e. isolation isn't sufficiently stressful to increase FSL in males, but isolation + swim stress is. From my reading, the authors did not test swim stress alone in either group. Conversely, the lack of FSL increase in response to swim stress in isolated females seems more like a ceiling effect than an occlusion. To be able to conclude that the isolation-induced FSL increase in females is due to stress, the authors should be able to show the same effect in group housed females exposed to swim stress.*

We apologize for the confusion. In order to further clarify things, we have now conducted the experiment suggested by the reviewer. Group-housed females were subjected to swim stress. FSL in CRH neurons was significantly longer (58.7 ± 1.9 ms, n=27) than FSL from unstressed, group-housed females (45.1 ± 1.9 ms, n=42, p=0.0001). In addition, we have subjected group housed males to swim. FSL was significantly longer (58.8 ± 2.8 ms, n=23) than FSL from unstressed, group-housed males (48.7 ± 2.1 ms, n=53, p<0.0001). These data are now included in a revised Figure 2.

*Additionally, Figure 2 shows the CORT/FSL correlation for males only, when the primary hypothesis seems to be for females. If this correlation wasn't significant in females, that should be stated. To me, these were the missing experiments and data sets in the first version of the paper and they appear to still be missing.*

The reviewer makes a good point. We focused on CORT immediately after swim in males because this give us a relatively confined temporal window in which to make the measurements. In the isolated females, although we can say that CORT is necessary (metyrapone experiment), we do not have the ability to monitor CORT levels in vivo in real time. In the absence of this information, it would be difficult to draw any conclusions from either a correlation or lack of correlation in this group.

*I can't find the data described at the end of the fourth paragraph of Results (CORT effects* in vitro*). First, why wouldn't the authors show these data, and second, why does it appear that this experiment was only done in isolated males? Response point 6 says that they didn't do it in females because FSL is already increased due to isolation, but they could have done it in non-isolated animals to show that CORT can have an effect on FSL in females. As with the first version of this manuscript, these kinds of decisions make it seem like the authors are throwing together a bunch of data without considering how each experiment contributes to the overarching conclusions.*

We have now conducted additional experiments in which slices from group-housed females were incubated with CORT. FSL in CRH neurons in this group was 64.1 ± 4.8 ms, n=32. This is significantly longer than FSL from non-CORT treated CRH neurons of group housed female mice (45.1 ± 1.9 ms, n=42, p<0.0001). These data are now included in a revised Figure 3.

*If the goal of this project is to define sexual dimorphism in CRH neuronal responses to stress and stress hormones, they need to do these experiments in both sexes and experimentally dissect out the roles of isolation vs. swim stress, since in many experiments these conditions are combined.*

We apologize for our lack of clarity. We have now conducted additional experiments in which group-housed males were also subjected to swim stress. This caused an increase in FSL. Importantly, the FSL in single males subjected to swim is not different than grouped males subjected to swim. These data are now included in a revised Figure 2.